# Persistent fatigue following SARS-CoV-2 infection is common and independent of severity of initial infection

**Liam Townsend**[1,2]☯*, **Adam H. Dyer**[3,4]☯, **Karen Jones**[3], **Jean Dunne**[3], **Aoife Mooney**[3], **Fiona Gaffney**[3], **Laura O'Connor**[3], **Deirdre Leavy**[3], **Kate O'Brien**[5], **Joanne Dowds**[5], **Jamie A. Sugrue**[6], **David Hopkins**[7], **Ignacio Martin-Loeches**[8], **Cliona Ni Cheallaigh**[1,2], **Parthiban Nadarajan**[9], **Anne Marie McLaughlin**[9], **Nollaig M. Bourke**[4], **Colm Bergin**[1,2], **Cliona O'Farrelly**[6,10], **Ciaran Bannan**[1,2‡], **Niall Conlon**[3,11‡]

**1** Department of Infectious Diseases, St James's Hospital, Dublin, Ireland, **2** Department of Clinical Medicine, School of Medicine, Trinity Translational Medicine Institute, Trinity College Dublin, Dublin, Ireland, **3** Department of Immunology, St James's Hospital, Dublin, Ireland, **4** Department of Medical Gerontology, School of Medicine, Trinity Translational Medicine Institute, Trinity College Dublin, Dublin, Ireland, **5** Department of Physiotherapy, St James's Hospital, Dublin, Ireland, **6** School of Biochemistry and Immunology, Trinity Biomedical Sciences Institute, Trinity College Dublin, Dublin, Ireland, **7** School of Medicine, Trinity College Dublin, Dublin, Ireland, **8** Department of Intensive Care Medicine, St James's Hospital, Dublin, Ireland, **9** Department of Respiratory Medicine, St James's Hospital, Dublin, Ireland, **10** Department of Comparative Immunology, School of Medicine, Trinity College Dublin, Dublin, Ireland, **11** Department of Immunology, School of Medicine, Trinity College Dublin, Dublin, Ireland

☯ These authors contributed equally to this work.
‡ CB and NC also contributed equally to this work.
* townsenl@tcd.ie

**Data Availability Statement:** All relevant data are within the manuscript.

**Funding:** LT has been awarded the Irish Clinical Academic Training (ICAT) Programme, supported

## Abstract

Fatigue is a common symptom in those presenting with symptomatic COVID-19 infection. However, it is unknown if COVID-19 results in persistent fatigue in those recovered from acute infection. We examined the prevalence of fatigue in individuals recovered from the acute phase of COVID-19 illness using the Chalder Fatigue Score (CFQ-11). We further examined potential predictors of fatigue following COVID-19 infection, evaluating indicators of COVID-19 severity, markers of peripheral immune activation and circulating pro-inflammatory cytokines. Of 128 participants (49.5 ± 15 years; 54% female), more than half reported persistent fatigue (67/128; 52.3%) at median of 10 weeks after initial COVID-19 symptoms. There was no association between COVID-19 severity (need for inpatient admission, supplemental oxygen or critical care) and fatigue following COVID-19. Additionally, there was no association between routine laboratory markers of inflammation and cell turnover (leukocyte, neutrophil or lymphocyte counts, neutrophil-to-lymphocyte ratio, lactate dehydrogenase, C-reactive protein) or pro-inflammatory molecules (IL-6 or sCD25) and fatigue post COVID-19. Female gender and those with a pre-existing diagnosis of depression/anxiety were over-represented in those with fatigue. Our findings demonstrate a significant burden of post-viral fatigue in individuals with previous SARS-CoV-2 infection after the acute phase of COVID-19 illness. This study highlights the importance of assessing those recovering from COVID-19 for symptoms of severe fatigue, irrespective of severity of initial illness, and may identify a group worthy of further study and early intervention.

by the Wellcome Trust and the Health Research
Board (Grant Number 203930/B/16/Z), the Health
Service Executive, National Doctors Training and
Planning and the Health and Social Care, Research
and Development Division, Northern Ireland
(https://icatprogramme.org/). NC is part-funded by
a Science Foundation Ireland (SFI) grant, Grant
Code 20/SPP/3685. The funders had no role in
study design, data collection and analysis, decision
to publish, or preparation of the manuscript.

**Competing interests:** The authors have declared
that no competing interests exist.

## Introduction

Fatigue is recognised as one of the most common presenting complaints in individuals
infected with SARS-CoV-2, the cause of the current COVID-19 pandemic. In early reports on
the clinical characteristics of those infected, fatigue was listed as a presenting complaint in 44–
69.6% [1–3]. Further studies were followed by meta-analyses, with 34–46% of those infected
presenting with fatigue [4–7]. Whilst the presenting features of SARS-CoV-2 infection have
been well-characterised, with symptoms ranging from mild taste and smell disturbance to dys-
pnoea and respiratory failure, the medium and long-term consequences of SARS-CoV-2 infec-
tion remain unexplored [1, 8, 9]. In particular, concern has been raised that SARS-CoV-2 has
the potential to trigger a post-viral fatigue syndrome [10, 11].

Patients acutely infected with SARS-CoV-2 demonstrate decreased lymphocyte counts,
higher leukocyte counts with an elevated neutrophil-to-lymphocyte ratio (NLR) in addition to
decreased percentages of monocytes, eosinophils and basophils. It has also been reported that
both helper and suppressor T cells are decreased in those with SARS-CoV-2 [12]. In severe
cases, elevated C-reactive protein (CRP), ferritin, d-dimers in addition to pro-inflammatory
factors such as IL-6 and soluble CD25 (sC25), and an increase in intermediate (CD16+ CD14
+) monocytes have been reported [13, 14]. Whether or not the immunological alterations seen
in SARS-CoV-2 have any relationship to the potential development of medium and long-term
symptoms following infection is an area which has not been researched to date. The persis-
tence of these changes following resolution of initial infection have also not been examined.

In one of the few reports to assess the long-term consequences of the severe acute respira-
tory syndrome (SARS) epidemic (caused by the novel coronavirus SARS-CoV), a subset of
patients in Toronto experienced persistent fatigue, diffuse myalgia, weakness and depression
one year after their acute illness and could not return to work [15]. In a similar follow-up
study amongst 233 SARS survivors in Hong Kong, over 40% of respondents reported a chronic
fatigue problem 40 months after infection [16]. In those affected by the subsequent Middle-
Eastern Respiratory Syndrome Coronavirus (MERS-CoV) outbreak, prolonged symptoms and
fatigue were reported up to 18 months after acute infection [17]. Similarly, prominent post-
infectious fatigue syndromes have been reported following Epstein-Barr Virus (EBV), Q-Fever
and Ross River Virus (RRV) infections, as well as rickettsiosis [18–22]. It has also been
reported following HHV-6 infection and in HIV infection [23, 24]. Whether or not infection
with SARS-CoV-2 has the potential to result in post-viral fatigue, both in the medium and
long-term, is currently unknown.

Persistent fatigue lasting 6 months or longer without an alternate explanation is termed
chronic fatigue syndrome (CFS). This may be observed after several viral and bacterial infec-
tions [11]. There have also been links between CFS and depression, although it remains
unclear whether one diagnosis precedes the onset of the other [25–27]. Whilst infections are
thought to precipitate CFS, the pathophysiology remains controversial. Studies of post-viral
fatigue and CFS often focus on immune system alterations, but robust data to indicate causa-
tion or association is absent. There are a plethora of studies examining immune dysregulation
and activation in CFS; however, none of these have provided a consistent finding or biologi-
cally plausible answer; rather, there are contrasting findings across studies concerning both
immune population changes and cytokine levels [28–30]. The heterogenous findings in
immune populations in CFS include changes in both lymphoid and myeloid populations [31–
34]. The disparate findings of prior CFS studies may be due to the variety of aetiologies that
ultimately lead to CFS. Whether alterations in immune system activity has any relationship to
the potential post-viral fatigue experienced with the novel SARS-CoV-2 is an important ques-
tion for future research. Prospectively examining patients following SARS-CoV-2 infection

provides a well characterised population with identical index infection, allowing for more accurate descriptors of both disease state and disease characteristics.

We sought to establish whether patients recovering from SARS-CoV-2 infection remained fatigued after their physical recovery, and to investigate whether there was a relationship between severe fatigue and a variety of clinicopathological parameters. We also sought to examine persistence of markers of disease beyond clinical resolution of infection.

## Materials and methods

### Study setting and participants

The current study was carried out in the post-COVID-19 review clinic at St James's Hospital (SJH), Dublin, Ireland. Participants were recruited from the post-COVID-19 outpatient clinic, which offers an outpatient appointment to all individuals with a positive SARS-CoV-2 naso-pharyngeal swab PCR at our institution. Patients attending the outpatient clinic were invited to participate in the current study by a research physician. In order to be considered for inclusion in the current study, participation had to occur at least 6 weeks after either: (i) date of last acute COVID-19 symptoms (for outpatients) and (ii) date of discharge for those who were admitted during their acute COVID-19 illness.

### Fatigue assessment

Fatigue was assessed using the validated Chalder Fatigue Scale (CFQ-11) [35, 36]. Briefly, participants are asked to answer these questions with particular reference to the past month in comparison to their pre-COVID-19 baseline, with responses measured on a Likert scale (0–3). From this a global score can be constructed out of a total of 33, as well as scores for the sub-scales of physical (0–21) and psychological (0–12) fatigue [37].

Further, the CFQ also allows the differentiation of "cases" vs "non-cases" where scores 0 and 1 ("Better than usual"/"No worse than usual") are scored a zero and scores 2 and 3 ("Worse than usual"/"Much worse than usual") are scored a 1 (bimodal scoring). The sum of all 11 binary scores is calculated and those with a total score of four or greater considered to meet the criteria for fatigue. This latter method for "caseness" is validated and closely resembles other fatigue questionnaires [37–40].

For the current study, we computed: (i) case-status (fatigue vs. non-fatigued) using the bimodal scoring method and the (ii) total CFQ-11 score (from a maximum of 33).

### Blood sampling & analysis of circulating pro-inflammatory cytokines

Blood sampling was incorporated as part of routine phlebotomy occurring on the same day as study participation/fatigue assessment. This involved measurement of routine laboratory parameters, including white cell counts (leukocyte, neutrophil and lymphocyte counts), CRP and lactate dehydrogenase (LDH). IL-6 and sCD25 levels were measured in serum by ELISA (R&D systems).

### Clinical covariate assessment

Routine demographic information was collected from participants. Further information was obtained from patient records and included: dates of COVID-19 symptoms, inpatient admission, treatment with supplemental oxygen and admission to the critical care/Intensive Care Unit (ICU). Background medical history was assessed by obtaining a list of regular medications and a list of medical comorbidities. A history of depression/anxiety was recorded as a physician-diagnosed history of depression/anxiety or regular use of antidepressant medication.

Additionally, participants were assessed for frailty which was operationalised using Rockwood's Clinical Frailty Scale (range 0–7) [41]. In order to assess subjective recovery from COVID-19 illness, participants were also asked a binary question regarding their perception of having returned to full health.

### Ethical approval

Informed written consent was obtained from all participants in the current study in accordance with the Declaration of Helsinki [42]. Ethical approval for the current study was obtained from the Tallaght University Hospital (TUH)/St James's Hospital (SJH) Joint Research Ethics Committee (reference REC 2020–04 (01)).

### Statistics

All statistical analysis was carried out using STATA v15.0 (Texas, USA) and statistical significance considered $p < 0.05$. Descriptive statistics are reported as means with standard deviations (SD) and median with interquartile ranges (IQR) as appropriate.

We analysed between-group differences in those with severe fatigue in comparison to those without severe fatigue (categorised as non-fatigued as per the case definition of the CFQ-11 above) using t-tests, chi-square tests and Wilcoxon rank-sum tests as appropriate (data were examined for normality using Q-q plots and histograms).

Logistic regression was used to analyse predictors of severe fatigue. We tested the association of severe fatigue with time interval between assessment and COVID-19 diagnosis, as well as several important indicators of COVID-19 severity (days since symptom onset, need for inpatient admission, supplemental oxygen treatment admission to critical care). These associations were tested individually using both an unadjusted model (model 1) and a model adjusted for age and sex (model 2). Subsequently, we analysed the associations between individual laboratory parameters/serum cytokines and severe fatigue using the same models. Results are presented as Odds Ratios (OR)/adjusted Odds Ratios (aOR) with corresponding 95% Confidence Intervals (CIs) and *p*-values.

Using the same independent variables and model adjustment, we examined the association between the above predictor variables and total CFQ-11 score in order to assess relationships not seen using the binary case definition. Linear regression was used testing each predictor individually (model 1) and again, adjusting for age and sex (model 2). Further exploratory analysis involved adding interaction terms with both age and gender, to examine for any potential gender or age-specific effects.

## Results

### Participant characteristics

223 patients were offered an outpatient appointment, of which 128 (57%) attended for assessment. The remaining 43% declined an outpatient appointment. These were consecutively enrolled (mean age: 49.5 ± 15 years; 53.9% female). Just over half (71/128; 55.5%) were admitted to hospital for treatment of COVID-19, with the remainder managed as outpatients (57/128; 44.5%). Of the admitted cohort, 35 (49%) received hydroxychloroquine, while 6 (8.5%) of these also received prednisolone. The remaining 36 inpatients did not receive any targeted therapy. No outpatients received targeted therapy. Administration of targeted therapy was at the discretion of the treating physician, in line with the guidelines in our institution at that time. Administration of COVID-directed therapy had no association with fatigue status. Just over half (66/128; 51.6%) were healthcare workers. Baseline characteristics are detailed in Table 1.

**Table 1. Baseline characteristics of study participants by fatigue case status ("caseness").**

| Characteristic | Overall (N = 128) | Non-Fatigued (N = 61) | Fatigued (N = 67) | Statistic |
|---|---|---|---|---|
| Age (mean ± SD) | 49.5 ± 15 | 49.7 ± 16 | 49.3 ± 14.3 | t = 0.16, p = 0.44 |
| Gender, female (N, %) | 69 (53.9%) | 24 (39.3%) | 45 (67.2%) | $\chi^2$ = 9.95, p = 0.002 |
| Body Mass Index, kg/m$^2$, (mean ± SD) | 28.7 ± 5.3 | 28.6 ± 4.9 | 28.8 ± 5.8 | t = -0.09, p = 0.54 |
| Clinical Frailty Scale (median, IQR) | 2 (1–2) | 2 (1–2) | 1 (1–2) | z = -0.15, p = 0.88 |
| Total Number of Medical Comorbidities (median, IQR) | 1 (0–2) | 1 (0–3) | 1 (0–2) | z = -1.40, p = 0.16 |
| Total Number of Regular Medications (median, IQR) | 1 (0–4) | 1 (0–4) | 0 (0–4) | z = -1.35, p = 0.18 |
| History of Anxiety/Depression | 10 (7.8%) | 1 (1.6%) | 9 (13.4%) | $\chi^2$ = 5.18, p = 0.02 |
| Interval from COVID-19 Symptoms to Fatigue Assessment | | | | |
| *<56 days (<8 weeks)* | 26 (20.3%) | 9 (14.8%) | 17 (25.4%) | |
| *56–69 days (8–10 weeks)* | 31 (24.2%) | 20 (32.8%) | 11 (16.4%) | |
| *69–83 days (10–12 weeks)* | 33 (25.8%) | 16 (26.2%) | 17 (25.4%) | |
| *>84 days (12 weeks)* | 38 (29.7%) | 16 (26.2%) | 22 (32.8%) | $\chi^2$ = 5.8, p = 0.12 |
| Total CFQ-11 Score (mean ± SD) [Liekert Scoring] | 15.8 ± 5.9 | 11.2 ± 3.2 | 20.0 ± 4.4 | t = -12.8, p<0.001 |
| Physical Fatigue (mean ± SD) [CFQ-11 items 1–7] | 11.38 ± 4.22 | 7.72 ± 1.87 | 14.54± 2.94 | z = -9.52, p<0.001 |
| Psychological Fatigue (mean ± SD) [CFQ-11 items 8–11] | 4.72 ± 1.99 | 3.79 ± 0.97 | 5.52 ± 2.29 | z = -5.91, p<0.001 |
| Total CFQ-11 Score (mean ± SD) [Bimodal Scoring] | 4.2 ± 3.5 | 1 ± 1.2 | 7 ± 2.2 | t = -18.6, p<0.001 |

SD: Standard Deviation, N: Number; IQR: Interquartile Range. CFQ-11: Chalder Fatigue Scale. Data are presented as means with standard deviations or medians with interquartile ranges as appropriate. Proportions are expressed both as numbers and percentages. Statistical analysis was carried out using t-tests, Wilcoxon rank sum tests and chi-square tests as appropriate in order to compare differences in those without fatigue and those non-fatigued/with non-severe fatigue as per the CFQ-11 "caseness" definition for severe fatigue.

The median interval between study assessment and discharge from hospital or a timepoint 14 days following diagnosis if managed as an outpatient was 72 days (IQR: 62–87). Fifty-four patients (54/128; 42.2%) reported feeling back to their full health, whilst the majority did not. Prior to COVID-19 illness, the majority (82%;105/128) had been employed, of whom 33 (31%) had not returned to work at time of study participation.

## Prevalence of post-COVID fatigue

Fatigue was assessed using the CFQ-11 in all participants and the mean (± SD) score was 15.8 ± 5.9 across the study population. The mean physical fatigue score (± SD) was 11.38 ± 4.22, while the mean psychological fatigue score (± SD) was 4.72 ± 1.99. Based on the CFQ-11 case definition, 52.3% (67/128) met the criteria for fatigue, with the mean (± SD) CFQ-11 score in this group being 20 ± 4.4. On univariate analysis of differences in those with and without fatigue, there was a greater number of females in addition to a greater number of participants with a history of anxiety/depression or anti-depressant use in the severe fatigue group ($\chi^2$ = 9.95, p = 0.002, $\chi^2$ = 5.18, p = 0.02 respectively), but no differences in other characteristics (Table 1). There was no association with being a healthcare worker and meeting the case definition for fatigue.

## COVID-19 disease characteristics and fatigue

Overall, there was no association, either using unadjusted models, or models adjusted for age and sex, between COVID-19 disease related characteristics (days since symptom onset, need for inpatient admission/supplemental oxygen/critical care, length of hospital stay) and either fatigue "caseness" (using logistic regression) or total CFQ-11 score (using linear regression) (See Table 2).

**Table 2.** Association of COVID-19, laboratory values and circulating pro-inflammatory cytokines with fatigue case status (Fatigue vs non-fatigued/non-severe fatigue) and total fatigue score (CFQ-11).

| | Non-Severe Fatigue (N = 61) | Severe Fatigue (N = 67) | Severe Fatigue (Logistic) | | | | CFQ-11 (Linear) | | | |
| --- | --- | --- | --- | --- | --- | --- | --- | --- | --- | --- |
| | | | Model 1 | | Model 2 | | Model 1 | | Model 2 | |
| Covid-19 Characteristics | | | OR (95% CI) | p | aOR (95% CI) | p | β (95% CI) | p | β (95% CI) | p |
| Days since Symptom Onset | 71 (66–85) | 73 (56–88) | 1.00 (0.98, 1.02) | 0.87 | 1.00 (0.98, 1.01) | 0.48 | -0.02 (-0.07, 0.03) | 0.48 | -0.04 (-0.08, 0.01) | 0.14 |
| Required Hospital Admission | 36 (59.0%) | 35 (52.2%) | 0.76 (0.38, 1.53) | 0.44 | 1.04 (0.44, 2.43) | 0.93 | -0.42 (-2.48, 1.66) | 0.57 | 0.96 (-1.29, 3.20) | 0.89 |
| Length of Stay, Days | 9.5 (6–19) | 8 (6–17) | 1.00 (0.95, 1.04) | 0.85 | 1.01 (0.95, 1.07) | 0.78 | -0.02 (-0.17, 0.14) | 0.85 | 0.03 (-0.12, 0.18) | 0.68 |
| Required Supplemental $O_2$ | 25 (41%) | 22 (32.8%) | 0.73 (0.34, 1.60) | 0.44 | 0.98 (0.18, 2.39) | 0.96 | -0.99 (-3.37, 1.40) | 0.41 | 0.21 (-2.20, 2.63) | 0.86 |
| Required Critical Care/ ICU | 10 (16.4%) | 8 (11.9%) | 0.55 (0.19, 1.52) | 0.24 | 0.82 (0.26, 2.56) | 0.73 | -2.90 (-6.09, 0.30) | 0.08 | -1.25 (-4.42, 1.92) | 0.44 |
| **Laboratory Values** | | | | | | | | | | |
| Leukocytes ($10^9$ cells/L) | 6.0 (5.3–7.2) | 6.3 (5.4–7.4) | 1.05 (0.87, 1.28) | 0.59 | 1.05 (0.86, 1.29) | 0.62 | 0.02 (-0.55, 0.58) | 0.96 | -0.02 (-0.55, 0.52) | 0.96 |
| Neutrophils ($10^9$ cells/L) | 3.2 (2.6–4.4) | 3.2 (2.8–4.3) | 1.09 (0.86, 1.39) | 0.49 | 1.08 (0.84, 1.40) | 0.53 | -0.01 (-0.71, 0.70) | 0.99 | -0.03 (-1.39, 1.33) | 0.97 |
| Lymphocytes ($10^9$ cells/L) | 2.0 (1.6–2.3) | 2.0 (1.6–2.5) | 1.00 (0.62, 1.61) | 0.99 | 0.91 (0.55, 1.50) | 0.72 | 0.29 (-1.15, 1.73) | 0.69 | 0.30 (-1.56, 2.16) | 0.75 |
| Neutrophil: Lymphocyte Ratio | 1.7 (1.2–2.3) | 1.6 (1.3–2.3) | 1.14 (0.89, 1.46) | 0.30 | 1.23 (0.95, 1.60) | 0.12 | -0.04 (-0.68, 0.61) | 0.81 | 0.18 (-0.44, 0.81) | 0.56 |
| LDH (U/L) | 185 (168–208) | 178 (165–195) | 1.00 (0.99, 1.01) | 0.69 | 1.00 (0.99, 1.01) | 0.50 | 0.01 (-0.02, 0.04) | 0.34 | 0.01 (-0.02, 0.04) | 0.52 |
| CRP (pg/mL) | 1.19 (0–2.52) | 1.68 (0–3.74) | 1.12 (0.99, 1.27) | 0.06 | 1.12 (0.99, 1.28) | 0.07 | 0.17 (-0.11, 0.44) | 0.23 | 0.12 (-0.12, 0.39) | 0.31 |
| IL-6 (pg/mL) | 0 (0–4.32) | 0 (0–3.52) | 0.90 (0.78, 1.03) | 0.13 | 0.90 (0.77, 1.06) | 0.21 | -0.18 (-0.54, 0.18) | 0.33 | -0.13 (-0.50, 0.25) | 0.50 |
| CD25 (pg/mL) | 1118 (883–1634) | 1137 (802–1606) | 1.00 (1.00, 1.00) | 0.76 | 1.00 (1.00, 1.00) | 0.18 | -0.00 (-0.00, 0.00) | 0.73 | 0.00 (-0.00, 0.00) | 0.41 |

CFQ-11: Chalder Fatigue Score; OR: Odds Ratio; CI: Confidence Interval; ICU: Intensive Care Unit; LDH: Lactate Dehydrogenase; CRP: C-Reactive Protein; IL-6: Interleukin-6; U/L: Units/Litre. Summary statistics are provided as medians with interquartile ranges or numbers with percentages as appropriate. Results of logistic regression are reported as OR and adjusted OR with appropriate 95% confidence intervals alongside corresponding p-values. Results of linear regressions are presented as Beta-coefficients β with appropriate 95% confidence intervals and p-values. Associations were tested unadjusted in the first instance (Model 1) with adjustment for Age and Gender (Model 2).

## Laboratory results and post-COVID-19 fatigue

The relationship between the values of six routine laboratory measures of inflammation and cell turnover (leukocyte, neutrophil and lymphocyte counts, NLR, LDH, CRP) had no relationship either to severe fatigue case-status (logistic regression) or total CFQ-11 score (linear regression) under either unadjusted models or those with adjustment for age and sex. Full results are reported in Table 2. There was similarly no association between the serum levels of IL-6 or soluble CD25 and either fatigue case-status or total CFQ-11 score. Of note, 112 participants (87.5%) had CRP levels within normal range (0–5 mg/L), 85/99 (85.9%) had IL-6 levels within the normal range (0–7.62 pg/mL) and 93/99 (93.9%) with soluble CD25 had levels within the normal range (0–2510 pg/mL).

## Discussion

The current study represents, to our knowledge, the first report in the literature examining the prevalence of fatigue following SARS-CoV-2 infection. There is a significant burden of fatigue at median follow up of 10 weeks, with half of the patient cohort reporting severe fatigue. This has several profound implications. Firstly, 50% of the participants do not feel back to full health, despite being medically deemed recovered from their primary illness. Secondly, the impact of this fatigue on daily function is already evident, with almost one third (31%) having not returned to employment. This is of particular concern, given that it is recommended that post-viral infection return to work should take place after four weeks to prevent deconditioning [43]. The large number of healthcare workers in our cohort is reflective of the overall demographics of Irish data and our institution, where 50% of positive SARS-CoV-2 cases involved healthcare workers [44]. The high proportion of healthcare workers infected by COVID-19, not just in our cohort but internationally, means that this will have a significant impact on healthcare systems [44–46]. These findings are independent of ageing effects, as age is known to be associated with increasing fatigue [47].

The rates of post-COVID fatigue appear much higher than those previously reported following EBV, Q fever or RRV infection at a similar interval [21]. However, post-SARS fatigue has been reported in 40% of individuals one year after initial infection, with 1 in 4 meeting CFS diagnostic criteria at that timepoint [16]. The levels of both physical and psychological fatigue seen post-COVID are higher than those of the general population, but do not reach the levels of those seen in chronic fatigue syndrome [48–50]. Rates of fatigue seen in our cohort are roughly equivalent to those reported in chronic disease states [51, 52]. Given that this cohort has no enduring evidence of active infection, the rate of fatigue is noteworthy. This is particularly important in relation to the 52% of the cohort that meet the diagnostic criteria for fatigue, as their CFQ-11 scores approach those seen in CFS cohorts [53, 54].

The findings concerning correlates of SARS-CoV-2-related fatigue are also notable. The absence of association with severity of initial infection has major implications on both the potential number of patients that may be affected and the burden this will place on healthcare services. Previous studies on SARS have generally focused on function and fatigue in post-ICU patients [55]. Our findings would suggest that all patients diagnosed with SARS-CoV-2 will require screening for fatigue. Our results also show a distinct female preponderance in the development of fatigue. This is in keeping with previous CFS findings [56]. We also noted significant association with pre-existing diagnosis of depression and use of anti-depressant medications and subsequent development of severe fatigue. While depression and CFS have previously been associated, there has been some debate as to the temporal relationship [21, 27]. Longitudinal studies will be needed to assess subsequent development of depression in the aftermath of post-COVID fatigue, as well as assessing the trajectory and persistence of fatigue.

The absence of a specific immune signature associated with persistent fatigue is a striking positive finding. As alluded to previously, CFS has been associated with a large number of differing changes in the inflammatory markers and immune cell populations. However, no consistent change has been reported across multiple studies [30]. This, in combination with our results, leads us to speculate that the pathological changes associated with CFS and post-COVID fatigue are more subtle. CFS may be the end point of a variety of distinct pathways or may be the consequence of pathological changes that are no longer systemically detectable. Despite a lack of distinct immunological findings, it is accepted that CFS can occur in the absence of demonstrable disease [57, 58]. The lack of distinct immune signature, coupled with the association with depression, lends credence to the multifactorial aetiology of CFS [59]. It

also supports the use of non-pharmacological interventions for fatigue management and provides no basis for the use of immunomodulation in treating post-COVID fatigue.

Our study concerned findings around COVID-19-related fatigue in the medium-term. In line with published data on viral dynamics, infectivity, and duration of infection, all our participants had recovered from their acute COVID-19 illness [60–62]. The median period between symptom onset and fatigue assessment was ten weeks, with no participant being recruited earlier than six weeks after their last COVID-19 symptoms or hospital discharge. Studies on CFS and post-viral fatigue have commonly assessed individuals at least 6 months after their viral illness. Post-SARS fatigue was described in 22 patients between 1 and 3 years post disease resolution; these patients were chosen due to their symptoms and may therefore not be representative of the overall cohort [15]. We feel that the short interval reported here is relevant due to the burden of fatigue seen and that COVID-19 patients were seen irrespective of post-disease symptoms, minimising the risk of selection bias. We also believe the effect fatigue has on self-perceived health and return to work is profound and worthy of reporting, especially in light of the number of patients that will be affected by this and the potential impact on individuals, employers and governments.

Management of fatigue states requires multi-disciplinary input and will not be appropriately addressed if follow-up is by treating medical physicians alone. A suite of interventions, including graded exercise and cognitive behavioural therapy, are needed to manage CFS and may be relevant to post infectious fatigue [63–65]. Furthermore, successful return to work will require ongoing input from occupational health departments and employers [66].

Our single centre study in a predominantly white Irish population has several limitations worthy of discussion. Our study is cross-sectional in nature and only assessed participants at a single timepoint. As previously mentioned, we are also reporting at a medium time point. As such, we would recommend that longitudinal studies are designed to assess patients at multiple time points and to examine the changes in immune markers and immune cell populations over time. It will also be illustrative to describe the persistence of fatigue at six months and beyond. It is important to note that there is no consensus on the nature of fatigue and its evaluation. However, the use of the widely applied Chalder Fatigue Scale is appropriate in this context. Further studies in large cohorts will be required to tease out fatigue subgroups and the potential complex factors at play. We also suggest that it is now time to consider the management of this post-COVID syndrome and advocate early analysis of multi-disciplinary fatigue management strategies.

## Conclusions

We present the first report, to our knowledge, of post-viral fatigue in those recovered from the acute phase of COVID-19 illness. In a similar fashion to previous coronavirus pandemics, COVID-19 appears to result in symptoms of severe fatigue that outlast the initial acute illness. Over half of individuals in the current study demonstrated symptoms consistent with severe fatigue a median of 10 weeks after their initial illness, while almost one-third of those previously employed had not returned to work. Most interestingly, fatigue was not associated with initial disease severity, and there were no detectable differences in pro-inflammatory cytokines or immune cell populations. Pre-existing diagnosis of depression is associated with severe post-COVID fatigue. This study highlights the burden of fatigue, the impact on return to work and the importance of following all patients diagnosed with COVID, not merely those who required hospitalisation. There are enormous numbers of patients recovering from SARS--CoV-2 infection worldwide. A lengthy post-infection fatigue burden will impair quality of life and will have significant impact on individuals, employers and healthcare systems. These

important early observations highlight an emerging issue. These findings should be used to inform management strategies for convalescent patients and allow intervention to occur in a timely manner.

## Acknowledgments

We would like to acknowledge the teams involved in the care of these patients, in particular the intensive care, immunology, psychological medicine and infectious diseases teams in St James's Hospital, as well as the contributions of the immunology diagnostic laboratory team. We would specifically like to acknowledge the contributions of the following people to the acquisition of data for this study: Dr Tara Kingston, Ms Patricia Byrne, Dr Katie Ridge, Dr Fionnuala Cox, Dr Catherine King.

## Author Contributions

**Conceptualization:** Liam Townsend, Ignacio Martin-Loeches, Cliona Ni Cheallaigh, Parthiban Nadarajan, Anne Marie McLaughlin, Nollaig M. Bourke, Colm Bergin, Cliona O'Farrelly, Ciaran Bannan, Niall Conlon.

**Data curation:** Liam Townsend, Adam H. Dyer, Deirdre Leavy.

**Formal analysis:** Liam Townsend, Adam H. Dyer.

**Funding acquisition:** Liam Townsend.

**Investigation:** Liam Townsend, Karen Jones, Jean Dunne, Aoife Mooney, Fiona Gaffney, Laura O'Connor, Kate O'Brien, Joanne Dowds, Jamie A. Sugrue, David Hopkins, Parthiban Nadarajan.

**Methodology:** Liam Townsend, Deirdre Leavy, Ciaran Bannan, Niall Conlon.

**Supervision:** Ignacio Martin-Loeches, Cliona Ni Cheallaigh, Nollaig M. Bourke, Colm Bergin, Cliona O'Farrelly, Ciaran Bannan, Niall Conlon.

**Validation:** Karen Jones, Jean Dunne, Aoife Mooney, Fiona Gaffney, Laura O'Connor, Deirdre Leavy.

**Writing – original draft:** Liam Townsend, Adam H. Dyer.

**Writing – review & editing:** Deirdre Leavy, Ignacio Martin-Loeches, Cliona Ni Cheallaigh, Parthiban Nadarajan, Anne Marie McLaughlin, Nollaig M. Bourke, Colm Bergin, Cliona O'Farrelly, Ciaran Bannan, Niall Conlon.

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
