## [Decision Letter · Decision Letter 0]

4 Sep 2020

PONE-D-20-22733

Re: Persistent fatigue following SARS-CoV-2 infection is common and independent of severity of initial infection

PLOS ONE

Dear Dr. Townsend,

Thank you for submitting your manuscript to PLOS ONE. After careful consideration, we feel that it has merit but does not fully meet PLOS ONE’s publication criteria as it currently stands. Therefore, we invite you to submit a revised version of the manuscript that addresses the points raised during the review process.

We look forward to receiving your revised manuscript.

Kind regards,

Giordano Madeddu

Academic Editor

PLOS ONE

Journal Requirements:

2. Please provide additional details regarding participant consent. In the ethics statement in the Methods and online submission information, please ensure that you have specified what type of consent you obtained (for instance, written or verbal, and if verbal, how it was documented and witnessed). If your study included minors, state whether you obtained consent from parents or guardians.

3. For studies involving humans categorized by race/ethnicity authors should update outmoded terms and potentially stigmatizing labels to more current, acceptable terminology e.g. “Caucasian” should be changed to “white” or “of [Western] European descent” (as appropriate).

Additional Editor Comments (if provided):

Dear authors,

I have received comments on your article from two independent reviewers. They suggest minor revisions to improve the manuscript.

however, I have noticed that the number of authors is excessive (26) and many of them do not follow the ICMJE criteria. The authors' list should be reduced to those fullfilling ICMJE criteria.

Best regards

Giordano Madeddu

Reviewers' comments:

Reviewer's Responses to Questions

**Comments to the Author**

1. Is the manuscript technically sound, and do the data support the conclusions?

Reviewer #1: Yes

Reviewer #2: Yes

2. Has the statistical analysis been performed appropriately and rigorously? 

Reviewer #1: Yes

Reviewer #2: Yes

3. Have the authors made all data underlying the findings in their manuscript fully available?

Reviewer #1: No

Reviewer #2: Yes

4. Is the manuscript presented in an intelligible fashion and written in standard English?

Reviewer #1: Yes

Reviewer #2: Yes

5. Review Comments to the Author

Reviewer #1: This is a very interesting paper on the pivotal issue of post-COVID 19 fatigue.

It is a very well written manuscript and I really appreciate the way it is structured and analyzed.

I have one question/request:

analysis and results = please provide and weight, if possible, any stratification of the timing of patients presentation >=6 weeks after acute infection, e.g. within 8 weeks, within 12 weeks etc.

Reviewer #2: Liam Townsend, with his coauthors, wrote an interesting paper about SARS-CoV-2 infection and persistent fatigue. The manuscript is well written and very interesting. However, some issues are present:

Introduction:

The introduction is well structured. I only suggest expanding the first part, adding the general symptomatology of COVID-19, both the major symptoms such as fever and dyspnea (10.26355/eurrev_202007_22291, https://doi.org/10.1016/S0140-6736(20)30183-5) and the minor such as anosmia and ageusia (https://doi.org/10.1002/hed.26269, https://doi.org/10.1002/hed.26204).

The authors wrote, “Similarly, prominent post-viral fatigue syndromes have been reported following Epstein-Barr Virus (EBV), Q-Fever and Ross River Virus (RRV) infections (16-19)." Q-fever is not caused by a virus. I suggest reformulating the sentence, and substitute ”post-viral fatigue ”with post-infection fatigue”, adding some other infectious disease such as Rickettsiosis, present in a recent review of the literature that should be added in the reference: https://doi.org/10.4084/mjhid.2020.056

Methods

The authors wrote, “Descriptive statistics are reported as means with standard deviations (SD) and interquartile ranges (IQR) as appropriate”. Please, modify the sentence with “Descriptive statistics are reported as means with standard deviations (SD) and median with interquartile ranges (IQR) as appropriate”.

I suggest to explicit the range of physical and psychological fatigue score (0 to …)

Results

I suggest deleting (See Table 1). At the end of the paragraph, the authors already wrote Baseline characteristics are detailed in Table 1.

The sentence “This is reflective of the overall demographics of Irish data and our institution, where 50% of positive SARS-CoV-2 cases involved healthcare workers (38).” should be moved in the Discussion.

It is not unclear, why the authors decide to adjust the regression model for age. Please comment.

No data are available about medical treatment. Do the authors know if these patients have been treated with hydroxychloroquine, steroids, or other drugs? Otherwise I suggest adding this lack in the limits of the study.

About the tables, it seems that some columns are missing; please fix it; otherwise, I will not be able to evaluate it.

Some percentage seems wrong. Please, recheck the tables (examples, Required Hospital Admission n people with severe fatigue: 35/67. The correct percentage is 52.2 and not 52.4; Required Critical Care/ICU in people with severe fatigue 8/67. The correct percentage is 11.9 and not 13.1).

Other issues

Some typos are present in the manuscript. Please recheck carefully and fix them.

In some reference the number of pages is missing.

6. PLOS authors have the option to publish the peer review history of their article (what does this mean?). If published, this will include your full peer review and any attached files.

Reviewer #1: **Yes: **Stefano Rusconi

Reviewer #2: No

---

## [Author Response · Author response to Decision Letter 0]

10 Sep 2020

Editorial Comments 

1. Style requirement: formatting, including font size and capitalization, now reflect PLOS One’s style 

2. Consent: clarification regarding written consent has been included

3. Race/ethnicity: the use of Caucasian has been replaced with White

4. Author number: the number of authors has been reduced in line with ICMJE criteria, with the addition of an acknowledgement section for those who contributed but do not meet authorship criteria 

Reviewer 1

1. Stratification regarding timing of follow up has been provided and added to Table 1

Reviewer 2

Introduction 

1. Additional information regarding the range of disease in acute infection has been added

2. Post-viral fatigue has been rephrased as post-infectious fatigue to reflect the presence of Q fever

Methods 

3. Minor correction to statistical methods made to include median 

4. The break-down range of physical and physiological fatigue is provided

Results 

5. Deletion of (See Table 1)

6. Sentence referring to burden of healthcare worker infection has been moved to discussion 

7. Rationale for controlling for age added to discussion, with reference; increasing age is known to be associated with increasing fatigue, hence it has been controlled for 

8. Information regarding the use of targeted therapy (hydroxychloroquine and corticosteroids) added to results 

9. Tables: Tables have been reformatted and percentages updated 

General Issues 

10. Article proofread and typos corrected

11. References checked and appropriate numbering and abbreviations added, in line with PLOS One’s requirements.

---

## [Decision Letter · Decision Letter 1]

24 Sep 2020

PONE-D-20-22733R1

Re: Persistent fatigue following SARS-CoV-2 infection is common and independent of severity of initial infection

PLOS ONE

Dear Dr. Townsend,

Thank you for submitting your manuscript to PLOS ONE. After careful consideration, we feel that it has merit but does not fully meet PLOS ONE’s publication criteria as it currently stands. Therefore, we invite you to submit a revised version of the manuscript that addresses the points raised during the review process.

We look forward to receiving your revised manuscript.

Kind regards,

Giordano Madeddu

Academic Editor

PLOS ONE

Reviewers' comments:

Reviewer's Responses to Questions

**Comments to the Author**

1. If the authors have adequately addressed your comments raised in a previous round of review and you feel that this manuscript is now acceptable for publication, you may indicate that here to bypass the “Comments to the Author” section, enter your conflict of interest statement in the “Confidential to Editor” section, and submit your "Accept" recommendation.

Reviewer #1: All comments have been addressed

Reviewer #2: All comments have been addressed

2. Is the manuscript technically sound, and do the data support the conclusions?

Reviewer #1: Yes

Reviewer #2: Yes

3. Has the statistical analysis been performed appropriately and rigorously? 

Reviewer #1: Yes

Reviewer #2: Yes

4. Have the authors made all data underlying the findings in their manuscript fully available?

Reviewer #1: Yes

Reviewer #2: Yes

5. Is the manuscript presented in an intelligible fashion and written in standard English?

Reviewer #1: Yes

Reviewer #2: Yes

6. Review Comments to the Author

Reviewer #1: (No Response)

Reviewer #2: Liam Townsend et al. re-submitted the manuscript, "Persistent fatigue following SARS-CoV-2 infection is common and independent of severity of initial infection". The authors have modified the article, as suggested in the previous revisions.

I have carefully reread the manuscript, and some issues are present.

The authors added that 41 patients received treatment with hydroxychloroquine. Have the authors checked if there are significant differences between the two groups (fatigued vs. non-fatigue)?.

Have the authors verified if people admitted to the hospital have were related to chronic fatigue?

In table one, the difference in BMI between the two groups had a p-value slightly above the cut-off of significance. Have the authors verified if having an obesity condition was associated with the development of chronic fatigue? Different studies about chronic fatigue in people with cancer showed how overweight people and, in particular, obese people had an increased risk to have chronic fatigue (https://doi.org/10.1007/s00520-019-04953-4, https://doi.org/10.1186/1471-2407-10-453)

In table one, the percentage should be rechecked. For example, the sum of the percentage in the row “Interval from COVID-19 Symptoms to Fatigue Assessment” about “people with fatigue” is 100.1%.

In table one, in the row “Interval from COVID-19 Symptoms to Fatigue Assessment”, the interval seems wrong because ten days are missing (56-69; 79-83)

In the subtitle of table one, the abbreviation “CFQ” should be explained.

In the subtitle of table two, the abbreviation “CI” “CRP” and “IL” should be explained. The abbreviation “OR” is reported two times.

There are some typos; please check all the manuscript; for example, in the discussion, the authors wrote “These finding are independent of ageing effects” instead of “These findings are independent of ageing effects”; “Given that this cohort have no enduring evidence of active infection” instead of “Given that this cohort has no enduring evidence of active infection”.

The word “multidisciplinary” one times was written with the hyphen, one time without it.

In the introduction, the authors changed “post-viral fatigue” with “post-infection fatigue” as suggested. The sentence now is “Similarly, prominent post-infectious fatigue syndromes have been reported following Epstein-Barr Virus (EBV), Q-Fever and Ross River Virus (RRV) infections.” In my opinion, three important infectious disease are missing: HIV infection (https://doi.org/10.1016/j.jpainsymman.2015.02.006), HHV-6 infection (https://doi.org/10.1155/1993/414602), and rickettsiosis (https://doi.org/10.4084/mjhid.2020.056). I suggest adding them.

7. PLOS authors have the option to publish the peer review history of their article (what does this mean?). If published, this will include your full peer review and any attached files.

Reviewer #1: **Yes: **Stefano Rusconi

Reviewer #2: No

---

## [Author Response · Author response to Decision Letter 1]

25 Sep 2020

Response to Reviewers

Dear Professor Madeddu and Editorial Board, 

Many thanks for the very helpful feedback. I have addressed each issue below, and have updated the manuscript accordingly. 

Editorial Comments 

N/A

Reviewer 1

N/A

Reviewer 2

1. The authors added that 41 patients received treatment with hydroxychloroquine. Have the authors checked if there are significant differences between the two groups (fatigued vs. non-fatigue)?.

Response: There was no difference in fatigue between those that received HCQ vs those that did not (17/35, 50% vs 18/36, 50%; p 0.99). A line stating this has been added to the manuscript. 

2. Have the authors verified if people admitted to the hospital have were related to chronic fatigue? 

Response: No patients in the cohort, admitted or otherwise, had a pre-existing fatigue syndrome. 

3. In table one, the difference in BMI between the two groups had a p-value slightly above the cut-off of significance. Have the authors verified if having an obesity condition was associated with the development of chronic fatigue? Different studies about chronic fatigue in people with cancer showed how overweight people and, in particular, obese people had an increased risk to have chronic fatigue (https://doi.org/10.1007/s00520-019-04953-4, https://doi.org/10.1186/1471-2407-10-453). 

Response: the p value for BMI is 0.54, well above the cut-off for significance. There is no association between BMI and fatigue state (p 0.41 under unadjusted conditions). 

4. In table one, the percentage should be rechecked. For example, the sum of the percentage in the row “Interval from COVID-19 Symptoms to Fatigue Assessment” about “people with fatigue” is 100.1%.

Response: Percentage for those attending at > 10 weeks corrected to 32.8%

5. In table one, in the row “Interval from COVID-19 Symptoms to Fatigue Assessment”, the interval seems wrong because ten days are missing (56-69; 79-83)

Rectified to 69 - 83

6. In the subtitle of table one, the abbreviation “CFQ” should be explained.

Response: legend amended 

7. In the subtitle of table two, the abbreviation “CI” “CRP” and “IL” should be explained. The abbreviation “OR” is reported two times.

Response: legend amended 

8. There are some typos; please check all the manuscript; for example, in the discussion, the authors wrote “These finding are independent of ageing effects” instead of “These findings are independent of ageing effects”; “Given that this cohort have no enduring evidence of active infection” instead of “Given that this cohort has no enduring evidence of active infection”.

Response: amended 

9. The word “multidisciplinary” one times was written with the hyphen, one time without it.

Response: amended 

10. In the introduction, the authors changed “post-viral fatigue” with “post-infection fatigue” as suggested. The sentence now is “Similarly, prominent post-infectious fatigue syndromes have been reported following Epstein-Barr Virus (EBV), Q-Fever and Ross River Virus (RRV) infections.” In my opinion, three important infectious disease are missing: HIV infection (https://doi.org/10.1016/j.jpainsymman.2015.02.006), HHV-6 infection (https://doi.org/10.1155/1993/414602), and rickettsiosis (https://doi.org/10.4084/mjhid.2020.056). I suggest adding them.

Response: Comments referring to these infections have been added. 

I hope we have satisfactorily addressed the issues raised. 

Yours sincerely

Liam Townsend

---

## [Editor Report · Decision Letter 2]

5 Oct 2020

Re: Persistent fatigue following SARS-CoV-2 infection is common and independent of severity of initial infection

PONE-D-20-22733R2

Dear Dr. Townsend,

We’re pleased to inform you that your manuscript has been judged scientifically suitable for publication and will be formally accepted for publication once it meets all outstanding technical requirements.

Kind regards,

Giordano Madeddu

Academic Editor

PLOS ONE
---

## [Editor Report · Acceptance letter]

29 Oct 2020

PONE-D-20-22733R2 

Persistent fatigue following SARS-CoV-2 infection is common and independent of severity of initial infection 

Dear Dr. Townsend:

I'm pleased to inform you that your manuscript has been deemed suitable for publication in PLOS ONE. Congratulations! Your manuscript is now with our production department. 

Kind regards, 

on behalf of

Dr. Giordano Madeddu 

Academic Editor

PLOS ONE